# The global distribution of the arbovirus vectors *Aedes aegypti* and *Ae. albopictus*

**Moritz UG Kraemer**[1]*, **Marianne E Sinka**[1], **Kirsten A Duda**[1], **Adrian QN Mylne**[2], **Freya M Shearer**[2], **Christopher M Barker**[3], **Chester G Moore**[4], **Roberta G Carvalho**[5], **Giovanini E Coelho**[5], **Wim Van Bortel**[6], **Guy Hendrickx**[7], **Francis Schaffner**[7], **Iqbal RF Elyazar**[8], **Hwa-Jen Teng**[9], **Oliver J Brady**[2], **Jane P Messina**[1], **David M Pigott**[1,2], **Thomas W Scott**[10,11], **David L Smith**[1,10,12], **GR William Wint**[13], **Nick Golding**[2], **Simon I Hay**[2,10,14]*

[1]Spatial Ecology and Epidemiology Group, Department of Zoology, University of Oxford, Oxford, United Kingdom; [2]Wellcome Trust Centre for Human Genetics, University of Oxford, Oxford, United Kingdom; [3]Department of Pathology, Microbiology, and Immunology, School of Veterinary Medicine, University of California, Davis, Davis, United States; [4]Department of Microbiology, Immunology and Pathology, Colorado State University, Fort Collins, United States; [5]National Dengue Control Program, Ministry of Health, Brasilia, Brazil; [6]European Centre for Disease Prevention and Control, Stockholm, Sweden; [7]Avia-GIS, Zoersel, Belgium; [8]Eijkman-Oxford Clinical Research Unit, Jakarta, Indonesia; [9]Center for Research, Diagnostics and Vaccine Development, Centers for Disease Control, Taipei, Taiwan; [10]Fogarty International Center, National Institutes of Health, Bethesda, United States; [11]Department of Entomology and Nematology, University of California, Davis, Davis, United States; [12]Sanaria Institute for Global Health and Tropical Medicine, Rockville, United States; [13]Environmental Research Group Oxford, Department of Zoology, University of Oxford, Oxford, United Kingdom; [14]Institute for Health Metrics and Evaluation, University of Washington, Seattle, United States

*For correspondence: moritz.
kraemer@zoo.ox.ac.uk (MUGK);
simon.hay@well.ox.ac.uk (SIH)

**Competing interests:**
See page 12

**Abstract** Dengue and chikungunya are increasing global public health concerns due to their rapid geographical spread and increasing disease burden. Knowledge of the contemporary distribution of their shared vectors, *Aedes aegypti* and *Aedes albopictus* remains incomplete and is complicated by an ongoing range expansion fuelled by increased global trade and travel. Mapping the global distribution of these vectors and the geographical determinants of their ranges is essential for public health planning. Here we compile the largest contemporary database for both species and pair it with relevant environmental variables predicting their global distribution. We show *Aedes* distributions to be the widest ever recorded; now extensive in all continents, including North America and Europe. These maps will help define the spatial limits of current autochthonous transmission of dengue and chikungunya viruses. It is only with this kind of rigorous entomological baseline that we can hope to project future health impacts of these viruses.

## Introduction

The mosquitoes *Aedes aegypti* [= *Stegomyia aegypti*] and *Aedes albopictus* [= *Stegomyia albopicta*] (*Reinert et al., 2009*) are vectors of several globally important arboviruses, including dengue virus (DENV) (*Simmons et al., 2012*), yellow fever virus (*Jentes et al., 2011*), and chikungunya virus (CHIKV) (*Leparc-Goffart et al., 2014*). The public health impact of DENV and CHIKV has increased

**eLife digest** Mosquitoes spread many disease-causing viruses and parasites between people and other animals, including viral infections such as dengue and chikungunya. Both infections cause high fevers often accompanied with excruciating joint pain or other flu-like symptoms. Dengue and chikungunya have become growing public health problems over the last fifty years. Today about half of the world's population is at risk of dengue infection, while chikungunya outbreaks, which were previously limited to Africa and Asia, have recently been reported in the Caribbean, South America and Europe.

The dengue and chikungunya viruses are transmitted between people by two species of mosquitoes called *Aedes aegypti* and *Ae. albopictus.* Therefore it is important to work out where these mosquito species are found around the globe to identify the areas at risk. It is also important to predict where these species could become established if they were introduced, in order to identify areas that could become at risk in the future.

Kraemer et al. now provide updated predictions about the distribution of these two mosquito species around the globe. These predictions are based upon the most up-to-date data on the known locations of the species combined with information on environmental conditions across the globe. The updated maps show that these *Aedes* mosquitoes are now found across all continents, including North America and Europe.

*Aedes albopictus* mosquitoes in particular are rapidly expanding their territory around the globe. Kraemer et al. used their new maps to show that, unlike in the United States, many of the areas in Europe and China that could support this mosquito species do not yet appear to have been colonized.

These findings provide a map of the distribution of both species as it stands at the moment. Further work is now needed to better understand which factors are contributing to the rapid expansion of these mosquitoes' range and what might be done to control this spread.

dramatically over the last 50 years, with both diseases spreading to new geographic locations and increasing in incidence within their range (*Weaver, 2014*). The remaining burden of vaccine-preventable yellow fever is similarly likely to be dramatically underestimated (*Garske et al., 2014*). DENV, with a nearly ubiquitous distribution in the tropics and more recently introduced to Europe (*ECDC, 2014*; *Schaffner and Mathis, 2014*), is the most prevalent human arboviral infection causing 100 million apparent annual infections world-wide with almost half of the world's population at risk of infection (*Brady et al., 2012*; *Bhatt et al., 2013*). CHIKV recently received considerable public health attention due to the outbreaks in Réunion in 2005–2006 (225,000 infections) (*Borgherini et al., 2007*), Italy in 2007 (205 infections) (*Rezza et al., 2007*), and France in 2010 and 2014 (2 and 11 locally transmitted cases, respectively) (*La Ruche et al., 2010*; *Grandadam et al., 2011*; *Paty et al., 2014*) as well as its recent invasion into the Americas with over 1 million cases recorded to date (*Cauchemez et al., 2014*; *Johansson et al., 2014*; *Morens and Fauci, 2014*). Increases in distribution and intensity of transmission are compounded by the lack of commercially available antivirals or vaccines for either disease (*Simmons et al., 2012*; *Roy et al., 2014*), although new therapeutics and vaccines are in development (*McArthur et al., 2013*; *Powers, 2014*; *Villar et al., 2015*). Similarly, while yellow fever infections have been on the decline due to extensive vector control and an effective vaccine developed more than 70 years ago, it still causes a significant disease burden in Africa and South America (*Poland et al., 1981*; *World Health Organization, 1990*; *Garske et al., 2014*). Given the public health impact of these diseases and their rapid global spread, understanding the current and future distribution, and determining the geographic limits of transmission and transmission intensity, will enable more efficient planning for disease control (*Carrington and Simmons, 2014*; *Semenza et al., 2014*; *Messina et al., 2015*). Because these diseases can only persist where their mosquito vectors, *Ae. aegypti* and *Ae. albopictus* are present, understanding the distributions of these two species underpins this strategy.

The global expansion of these arboviruses was preceded by the global spread of their vectors (*Charrel et al., 2014*). *Ae. aegypti* originated in Africa where its ancestral form was a zoophilic treehole mosquito named *Ae. aegypti formosus* (*Brown et al., 2014*). The domestic form *Ae. aegypti*

is genetically distinct with discrete geographic niches (*Brown et al., 2011*). It was hypothesised that due to harsh conditions coupled with the onset of the slave trade, *Ae. aegypti* were introduced into the New World from Africa, from where it subsequently spread globally to tropical and sub-tropical regions of the world (*Brown et al., 2014*). *Ae. albopictus,* originally a zoophilic forest species from Asia, spread to islands in the Indian and Pacific Oceans (*Delatte et al., 2009*). During the 1980s it rapidly expanded its range to Europe, the United States and Brazil (*Medlock et al., 2012*; *Carvalho et al., 2014*). Today both *Ae. aegypti* and *Ae. albopictus* are present in most Asian cities and large parts of the Americas (*Lambrechts et al., 2011*). *Ae. aegypti* feed almost exclusively on humans in daylight hours and typically rest indoors (*Scott and Takken, 2012*). In contrast *Ae. albopictus* is usually exophagic and bites humans and animals opportunistically (*Paupy et al., 2009*) but has also been shown to exhibit strongly anthropophilic behavior similar to *Ae. aegypti* in specific contexts (*Ponlawat and Harrington, 2005*; *Delatte et al., 2010*).

A number of previous studies have mapped the global or regional distributions of *Ae. aegypti* and *Ae. albopictus* by focusing on different aspects of their ecology. The majority examined the impacts of climatic conditions, often with an exclusive focus on temperature. *Kobayashi et al. (2002)* and *Nawrocki and Hawley (1987)* used results from laboratory studies to identify potential limits of establishment in Japan and Asia suggesting a minimum mean temperature in the coldest months of −2℃ and −5℃ respectively limits their distribution. *Brady et al. (2013)* extended that work by modeling the adult survival of both species under laboratory and field conditions, indicating that *Ae. albopictus* has higher survival rates than *Ae. aegypti*, though adults of the latter can tolerate a wider range of temperatures. Applying these results to global temperature data, *Brady et al. (2014)* produced maps indicating areas where the temperature is suitable for these vectors to persist. Whilst temperature is clearly a crucial factor constraining the distribution of the two species, these results alone are not sufficient to discriminate between areas where the species can and cannot persist. Other studies went further using statistical models, predicting the distributions of both species (though particularly *Ae. albopictus*) using a broader range of climatic variables including precipitation (*Benedict et al., 2007*; *Medley, 2010*; *Fischer et al., 2011*; *Caminade et al., 2012*; *Khormi and Kumar 2014*; *Campbell et al., 2015*).

Whilst these studies incorporated several generic climatic factors to predict the current and future distribution of the species, we were able to integrate a bespoke species-specific temperature suitability covariate and account for anthropogenic factors that are known to influence *Ae. aegypti* and *Ae. albopictus* distributions (*Reiter et al., 2003*). Both species are container-inhabiting but differ in their behaviour and biology so that they occupy different niches (*Eisen and Moore, 2013*). A few local studies showed, however, that local spread of *Ae. albopictus* and declining *Ae. aegypti* populations might be linked to inter-species competition (*O'Meara et al., 1995*; *Daugherty et al., 2000*; *Juliano et al., 2007*) and/or non-reciprocal cross-species inseminations (*Bargielowski et al., 2013*). Socio-economic factors affecting the distribution of the *Aedes* mosquitoes other than the use of containers to store water, include the use of air-conditioning, housing quality, and the rate of urbanisation (*Ramos et al., 2008*; *Aström et al., 2012*). In addition to exclusively focusing on meteorological factors in determining the spatial extent of the *Aedes* mosquitoes, many models used small sets of input occurrence data, which were biased towards particular countries with well-developed surveillance systems, such as, Brazil and Taiwan (*Benedict et al., 2007*; *Medley, 2010*; *Fischer et al., 2011*; *Campbell et al., 2015*).

In this context, we set out to model the global distribution of these two important vector species, compiling the most comprehensive occurrence dataset to date from published literature and national entomological surveys. To overcome previous modelling limitations, a probabilistic species distribution model using Boosted Regression Trees (BRT) was produced for each vector. Our models combine environmental and, for the first time, land-cover variables to predict the global distribution of both species at high spatial resolution. Importantly, the models quantify prediction uncertainty and aim at identifying key contributing factors and inter-species differences in their environmental niches.

## Results

In total, data collection yielded 19,930 and 22,137 spatially unique occurrence records for *Ae. aegypti* and *Ae. albopictus* respectively, which were used to train the distribution models. This includes up-to date records from national entomological surveys from Brazil and Taiwan for both species (*Carvalho et al., 2014*; *Yang et al., 2014*). For *Ae. aegypti,* >60% of all occurrence records are from Asia and

Oceania, 35% are from the Americas and only 575 unique occurrences are available for Africa and Europe (*Table 1a*). Similarly for *Ae. albopictus,* most of the occurrences are from Asia (75%), 23% are from the Americas and only 542 records are available from Europe and Africa (*Table 1b*). For each continent the top 10 countries in terms of occurrences recorded are shown for both species (*Table 1*). The geographic distribution of the occurrence records is the widest ever recorded with particularly high spatial and temporal resolution in Taiwan and Brazil for both species and in the United States for *Ae. albopictus*. All occurrence data have been made openly available through an online data repository to ensure consistency and reproducibility (*Pigott and Kraemer, 2014*; *Kraemer et al., 2015a*).

Maps showing the predicted global distribution for *Ae. aegypti* and *Ae. albopictus* are presented in *Figures 1*, *2*, respectively. The distributions of the two species differ markedly in a number of places. *Ae. aegypti* is predicted to occur primarily in the tropics and sub-tropics, with concentrations in northern Brazil and southeast Asia including all of India, but with relatively few areas of suitability in Europe (only Spain and Greece) and temperate North America. In Australia, however, *Ae. aegypti* shows a wider geographic distribution than *Ae. albopictus,* which is confined to the east coast, largely reflecting the known historic distribution of *Ae. aegypti*. By contrast, the distribution of *Ae. albopictus* extends into southern Europe (*Figure 3A*), northern China, southern Brazil, northern United States (3b), and Japan. Again, this reflects the current and historic distribution of *Ae. albopictus* and the ability of the species to tolerate lower temperatures (*Tsuda and Takagi, 2001*; *Lounibos et al., 2002*; *Thomas et al., 2012*; *Brady et al., 2014*).

**Table 1**. The geographic distribution of spatially unique occurrence records for the Americas, Europe/Africa, and Asia/Oceania

| | Country | Occurrences | | Country | Occurrences | | Country | Occurrences |
|---|---|---|---|---|---|---|---|---|
| *Ae. aegypti* | | | | | | | | |
| Americas | Brazil | 5,044 | Europe/Africa | Senegal | 112 | Asia/Oceania | Taiwan | 9,490 |
| | USA | 436 | | Cameroon | 55 | | Indonesia | 603 |
| | Mexico | 411 | | Kenya | 52 | | Thailand | 495 |
| | Cuba | 177 | | United Republic of Tanzania | 44 | | India | 423 |
| | Argentina | 170 | | Côte d'Ivoire | 40 | | Australia | 282 |
| | Trinidad and Tobago | 152 | | Nigeria | 35 | | Viet Nam | 223 |
| | Venezuela | 130 | | Madagascar | 28 | | Malaysia | 112 |
| | Colombia | 128 | | Gabon | 27 | | Singapore | 44 |
| | Puerto Rico | 120 | | Mayotte | 20 | | Philippines | 36 |
| | Peru | 89 | | Sierra Leone | 20 | | Cambodia | 29 |
| *Ae. albopictus* | | | | | | | | |
| Americas | Brazil | 3,441 | Europe/Africa | Italy | 203 | Asia/Oceania | Taiwan | 15,339 |
| | USA | 1,594 | | Madagascar | 58 | | Malaysia | 186 |
| | Mexico | 50 | | Cameroon | 42 | | Indonesia | 161 |
| | Cayman Islands | 15 | | France | 37 | | India | 150 |
| | Haiti | 13 | | Gabon | 27 | | Japan | 97 |
| | Guatemala | 12 | | Albania | 22 | | Thailand | 82 |
| | Venezuela | 7 | | Mayotte | 21 | | Singapore | 44 |
| | Colombia | 3 | | Greece | 18 | | Lao People's Democratic Republic | 26 |
| | Cuba | 3 | | Israel | 17 | | Philippines | 22 |
| | Puerto Rico | 3 | | Lebanon | 15 | | Viet Nam | 18 |

Top 10 countries in terms of occurrence records for each continent are shown for *Ae. aegypti* (a) and *Ae. albopictus* (b).

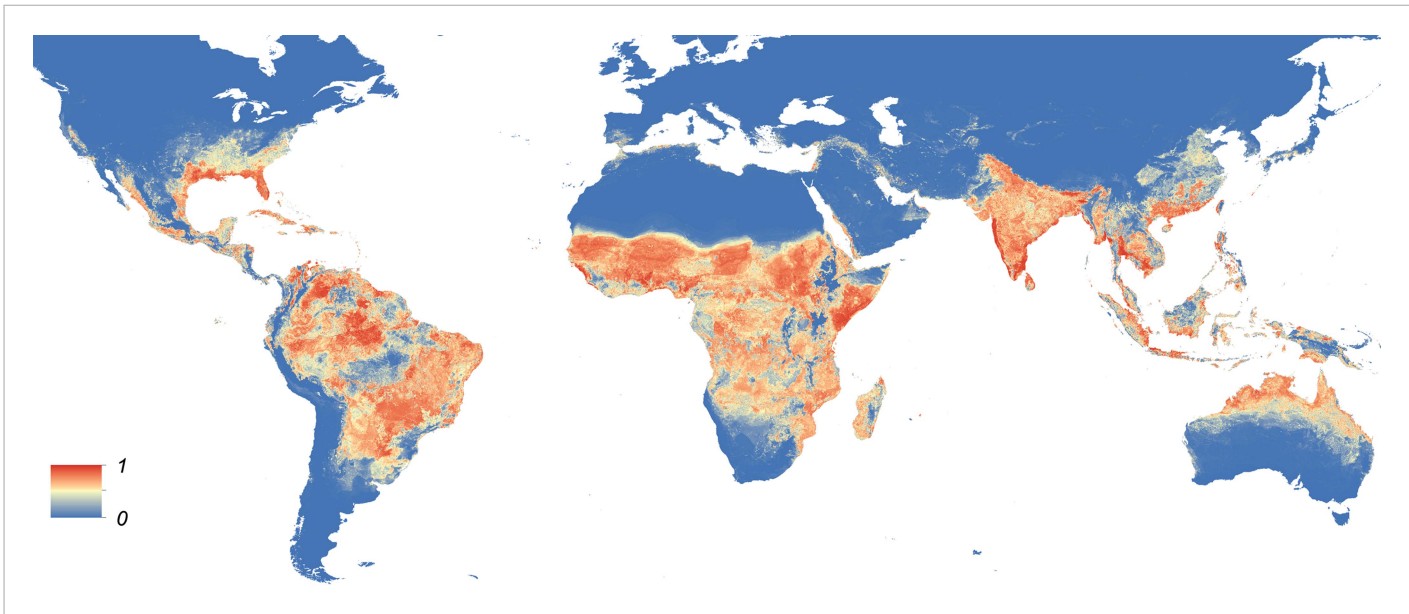

**Figure 1**. Global map of the predicted distribution of *Ae. aegypti*. The map depicts the probability of occurrence (from 0 blue to 1 red) at a spatial resolution of 5 km × 5 km.

The following figure supplements are available for figure 1:

**Figure supplement 1**. Effect plots of covariates used in this study showing the marginal effect of each covariate on probability of presence for *Ae. aegypti* (1) and *Ae. albopictus* (2): enhanced vegetation index (EVI) annual mean (**A**); Enhanced vegetation index—range (**B**); annual monthly maximum precipitation (**C**); annual monthly minimum precipitation (**D**); temperature suitability (**E**); urban areas (**F**); peri-urban areas (**G**).

**Figure supplement 2**. Set of covariate layers used to predict the ecological niche of *Ae. aegypti* and *Ae. albopictus* described in detail in the 'Materials and methods' section; (**A**) enhanced vegetation index (EVI) annual mean, (**B**) EVI annual range, (**C**) annual monthly maximum precipitation, (**D**) annual monthly minimum precipitation, (**E**) temperature suitability for *Ae. albopictus*, (**F**) temperature suitability for *Ae. aegypti*, (**G**) rural, peri-urban and urban classification layer.

**Figure supplement 3**. Visualization of pixel level uncertainty calculated using the upper and lower bounds of the 95% confidence intervals associated with the prediction maps for *Ae. aegypti* (**A**) and *Ae. albopictus* (**B**).

**Figure supplement 4**. The distribution of the occurrence database for *Ae. aegypti* (**A**) and *Ae. albopictus* (**B**) plotted on the underlying prediction surface.

In Europe, the predicted potential distribution of *Ae. albopictus* contains most of the known occurrence points, but suitability is also predicted in Portugal and the west of Spain, and in much of south-eastern Europe and the Balkans, where the species has yet to be reported. Similarly, in China *Ae. albopictus* has yet to be reported from much of the area predicted to be environmentally suitable. By contrast, in the United States the species has been reported from almost all of the predicted suitable areas, with the exception of a small band of predicted suitability on the western slope of the Sierra Nevada. Due to the relatively sparse reporting from Africa it remains uncertain whether areas predicted to be highly suitable are already infested or have yet to be colonized by the species. *Ae. albopictus* for example has only been reported from some West African countries (Nigeria, Cameroon, Gabon, the Central African Republic, Congo, Côte d'Ivoire) and Madagascar, and South Africa (as well as some islands in the Indian Ocean). The distribution of *Ae. aegypti* in Africa seems to be much wider, with reports of species occurrence in over 30 countries.

For both species, the most important predictor was temperature. Temperature suitability indices had high relative influence statistics for both species; this variable was selected in approximately half of regression tree decisions for *Ae. aegypti* (54.9%, CI = 53.7–56%) and *Ae. albopictus* (44.3%, CI = 42.7–45.6%). The full definition of a relative influence statistic is given in the 'Materials and methods' section under the heading *Predictive performance and relative influence of covariates*.

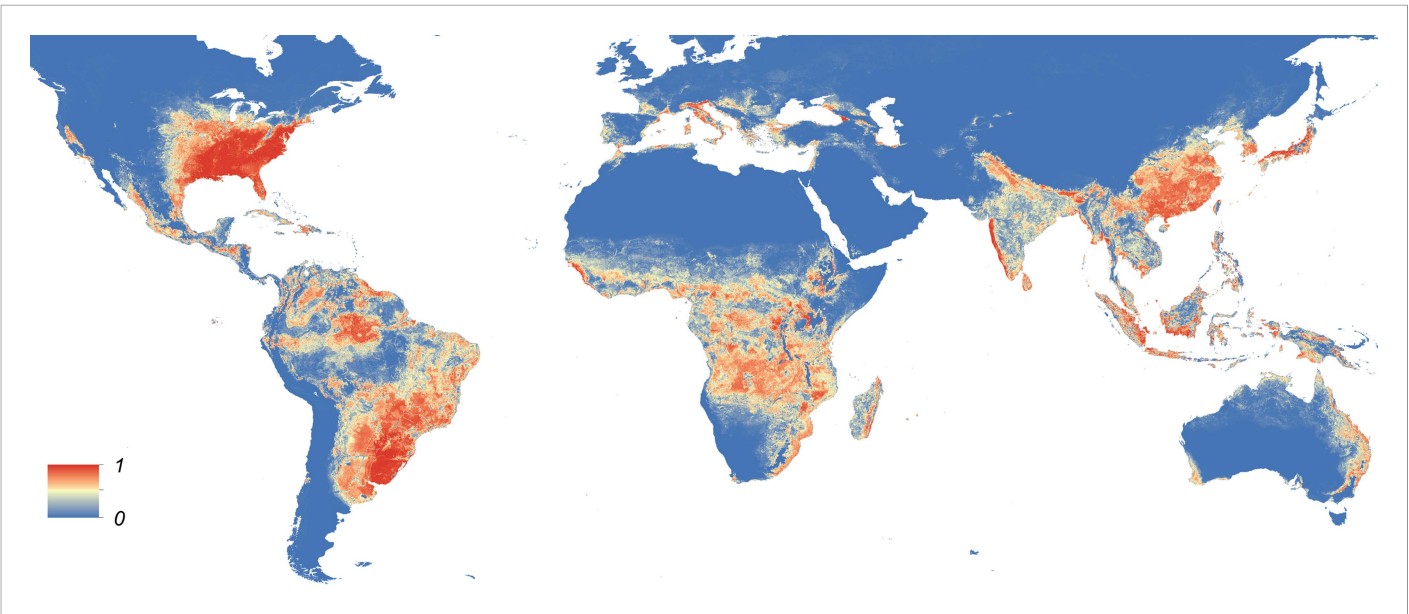

**Figure 2**. Global map of the predicted distribution of *Ae. albopictus*. The map depicts the probability of occurrence (from 0 blue to 1 red) at a spatial resolution of 5 km × 5 km.

Precipitation and vegetation indices made up the remainder of predictors. Urban land cover made very little contribution to either model (*Table 2*). Model evaluation statistics under cross-validation were high (AUC: 0.87 and 0.9 respectively) for both model ensembles, indicating high predictive performance of the model. Effect plots for each covariate are shown in *Figure 1—figure supplement 2*. Maps of uncertainty associated with these predictions are presented in *Figure 1—figure supplement 3*.

## Discussion

By combining the most comprehensive dataset of occurrence records with an advanced modelling approach and a bespoke set of environmental and land-cover correlates, we have produced contemporary high-resolution probability of occurrence maps for *Ae. aegypti* and *Ae. albopictus*, two of the most important disease vectors globally. Dengue and chikungunya, pathogens transmitted by these vectors and rapidly expanding in their distributions, are increasingly prominent in public health agendas and pose significant health threats to humans (*Staples et al., 2009*; *Gardner et al., 2012*; *Bhatt et al., 2013*; *Weaver and Lecuit, 2015*). In common with previous work to map the global distributions of the dominant vectors of malaria (*Sinka et al., 2010a*, *2010b*, *2011*), the maps will improve efforts to understand the spatial epidemiology of associated arboviruses, and to predict how these could change in the future. Specifically, these maps may be used to prioritize surveillance for these vector species and the diseases caused by the viruses they transmit in areas where disease and entomological reporting remains poor. For example, in parts of Asia and Africa where there is a mismatch between predicted environmental suitability and reported occurrences, these maps could be used to determine whether the vector has yet to fill its niche or if it is present but has not been reported due to limited entomological surveillance. They may also be used to identify areas where the species could persist but has yet to be reported, in order to proactively prevent vector establishment.

The relative contributions of each of the environmental covariates to the global models concur with our theoretical and experimental understanding of each species' biology. Both species' distributions are highly dependent on the limiting factor temperature places on survival of the adult mosquitoes and on the gonotrophic cycle (*Brady et al., 2013*) (*Table 2*). The inclusion of a bespoke temperature suitability index (*Brady et al., 2014*), both in defining the pseudo-absences and as a covariate, allowed us to capture both geographic and temporal variations in the species-specific effects of temperature in a single variable, leading to improved predictive skill of the models. As both

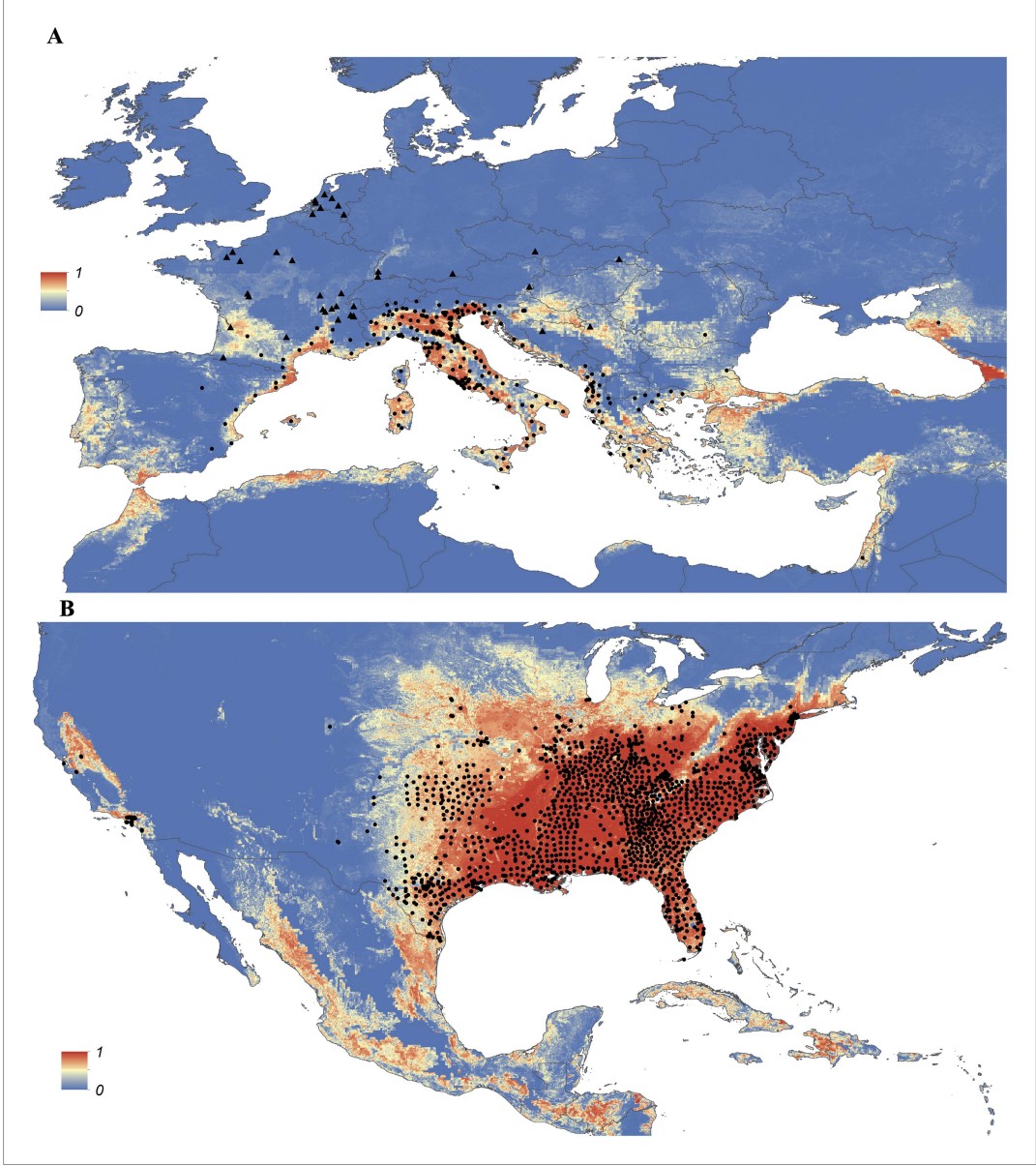

**Figure 3**. Predicted probability of occurrence of *Ae. albopictus* in Europe (**A**) and the United States (**B**), regions in which *Ae. albopictus* is rapidly expanding its range. Points represent known occurrences (transient [triangles] or established [circles]) until the end of 2013.

*Ae. aegypti* and *Ae. albopictus* lay their eggs in small water-filled containers (*Morrison et al., 2004*), it is encouraging that precipitation also has a strong influence on the model's predictions. The stronger influence of minimum precipitation for *Ae. albopictus* than for *Ae. aegypti* (16.1% vs 9.1%, *Table 2*) may reflect the former species' preference for non-domestic juvenile habitats, which are solely reliant on filling via precipitation. By contrast, *Ae. aegypti* primarily inhabits domestic water-holding containers (*Scott et al., 2000*) that are maintained in low-precipitation environments by water storage activities. The greater importance of enhanced vegetation index (EVI) for *Ae. albopictus* than for *Ae. aegypti* (15.3% vs 12.1%, *Table 2*) also supports the hypothesis that *Ae. albopictus* tends to prefer non-domestic juvenile sites (*Morrison et al., 2004*). This does not, however, rule out the possibility that the two species can overlap. Additional finer scale studies need to be conducted to investigate if competitive exclusion for hosts and/or habitat occurs between *Ae. aegypti* and *Ae. albopictus*. The effect of urbanicity was surprisingly low for both species (2% and 1.1% for *Ae. albopictus* and

**Table 2**. Relative contribution of environmental covariates predicting the global distribution of *Ae. aegypti* and *Ae. albopictus*

| | Mean contribution Ae. aegypti (%) | 95% confidence interval Ae. aegypti (%) | Mean contribution Ae. albopictus (%) | 95% confidence interval Ae. albopictus (%) |
|---|---|---|---|---|
| Temperature suitability | 54.9 | 53.7–56 | 44.3 | 42.7–45.6 |
| Maximum precipitation | 13.6 | 12.6–14.6 | 13.9 | 12.7–14.9 |
| Enhanced vegetation index (mean) | 12.1 | 11.3–12.9 | 15.3 | 14.5–16.3 |
| Minimum precipitation | 9.1 | 8.5–10 | 16.1 | 15.2–16.9 |
| Enhanced vegetation index (range) | 8.3 | 7.7–9 | 9.1 | 8.3–10.1 |
| Urbanicity | 2 | 1.3–2.4 | 1.1 | 0.7–1.7 |

*Ae. aegypti,* respectively). As both species have been shown to inhabit a wide variety of urban and peri-urban settings with various degrees of intensity (*Powell and Tabachnick, 2013*; *Li et al., 2014*), it is likely that the simple urban/rural distinction of our urbanicity covariate did not sufficiently capture this variation and instead continuous covariates such as EVI allow to better distinguish the respective habitat types and were thus chosen more frequently by the model. Incorporating a larger set of covariates allowed us to investigate not only the effect of temperature on survival but for additional variance as shown in the relative influence plots (*Figure 1—figure supplement 1*). Future *Aedes* species distribution models could be improved by including a comprehensive global covariate that distinguishes human settlements using complex satellite imagery processing tools (*Schneider, 2012*).

Our maps are based on covariates where each 5 km × 5 km pixel represents yearly mean average values. We therefore produce maps that represent the long-term average distribution of both species. However, this does not allow us to directly infer seasonal patterns of distributions which might be of importance on the periphery of the species distributions. With a more temporally resolved dataset it may be possible to capture the effects of intra-annual seasonality on the species' distributions. Adding mechanistic determinants, such as survival, have previously been used to combine seasonal patterns with global distribution maps (*Johansson et al., 2014*). To make best use of the comprehensive set of data collected, we construct models and maps at a global scale, allowing the model to share information across the whole spectrum of environmental regions. However, given the scale at which this study was performed, there is always the possibility that variation in microclimate or local adaptive strategies of both species may have a significant impact in some locations.

Previous studies have discussed the risk of pathogen importation and autochthonous transmission of DENV and CHIKV in Europe and the Americas without comprehensively accounting for the distribution of the vectors (*Bogoch et al., 2014*; *Schaffner and Mathis, 2014*). These freely available vector distributions maps (http://goo.gl/Zl2P7J) can now be used as covariates to refine these studies and to generate high-resolution maps of the risk of possible local DENV and CHIKV transmission in currently non-endemic settings. Such maps would be useful for prioritizing surveillance in areas where there is a risk of disease importation. This will be especially important in areas where sporadic cases of related viruses have been reported, such as Europe, the United States, Argentina, and China (*Rezza et al., 2007*; *Otero and Solari, 2010*; *Wu et al., 2010*; *Johansson, 2015*).

Both *Ae. aegypti* and *Ae. albopictus* have a history of global expansion associated with trade and travel (*Tatem et al., 2006*; *Brown et al., 2014*; *Gloria-Soria et al., 2014*). Introductions of the species over long distances and between continents has been associated with international trade routes via shipping and overland spread driven by human movement and transport routes, both facilitated by the endophilic behavior of the two species (*Nawrocki and Hawley, 1987*; *Tatem et al., 2006*; *Hofhuis et al., 2009*). The global spread of the associated pathogens has undoubtedly been a consequence of increasing global connectedness. As these processes continue and the world becomes increasingly connected and urbanized, risk of importation and subsequent autochthonous transmission of DENV and CHIKV will continue to increase (*Allwinn et al., 2008*; *Tomasello and Schlagenhauf, 2013*; *Khan et al., 2014*; *Messina et al., 2015*). The true distribution of both species is influenced by a variety of factors, not just the ones presented here. Nevertheless, this study represents an important baseline for further refinements. For instance, our maps can be used to indicate areas

where the species are likely to become established if introduced. Accurately predicting the future distributions of these species will also require model-based estimates of the rate at which these species colonize new areas. Such predictions can be informed by human and trade mobility patterns between endemic and non-endemic regions as well as data on the past spread of the vectors. Improving our ability to predict rates of vector importation will therefore be crucial to inferring future risk (*Seebens et al., 2013*).

Previous studies have provided crucial information on genetic variation both within and between populations of these two vector species (*Brown et al., 2011*). As the volume of georeferenced information on the population genetics of *Ae. aegypti* and *Ae. albopictus* increases, the potential to incorporate this information into mapping analyses to understand the current and future distribution of disease risk also increases. Phylogeographic analyses offer a unique way to infer the recent patterns of vector spread and to identify the major routes of importation (*Allicock et al., 2012*). This information is crucial to inform models that predict the risk of vector introductions.

Phylogenetic information could also be used to inform future iterations of the species distribution models used here by enabling the model to characterize and map environmental suitability for different vector subspecies. This could be particularly useful in the case of *Ae. albopictus* where genetic variation is known to underlie the ability to undergo diapause and therefore to overwinter in colder locations (*Takumi et al., 2009*). Mapping the distributions of distinct genetic subgroups could also improve our understanding of the complex interactions between mosquito vector populations and virus strains and how this relates to spatial variation in transmission intensity (*Tsetsarkin et al., 2007*; *Vazeille et al., 2007*; *Tsetsarkin and Weaver, 2011*; *Zouache et al., 2014*).

The maps presented comprise a contemporary estimate of the current and potential future distribution of *Ae. aegypti* and *Ae. albopictus*. As more occurrence data become available, these maps can be refined to incorporate recent importation and establishment events and corresponding improvements in predictions. By disseminating both the occurrence data and the predictive maps on an open-access basis we hope to facilitate both the future development of these maps and their uptake by the global public health community.

## Materials and methods

A BRT modelling approach was applied to derive probabilistic global environmental risk maps for *Ae. aegypti* and *Ae. albopictus*. BRT models are machine-learning model ensembles commonly used in species distribution modelling (SDM) and show strong predictive performance due to their ability to handle complex non-linear relationships between probability of species occurrence and multiple environmental correlates (*Elith et al., 2006*, *2008*). Our model required the following sets of input data in order to make accurate predictions of the distribution of these two species: (i) a temperature suitability mask defining the fundamental limits of both species; (ii) a globally comprehensive dataset of geo-positioned occurrence points for both species; (iii) appropriate land-cover and environmental covariate datasets that help explain the current distribution of the species; and (iv) a set of species absence records that further refine the species range and reduce sampling bias. Details regarding the specific attributes of the model and data generation are outlined below and maps of each of the covariates are shown in *Figure 1—figure supplement 2*.

### Temperature suitability mask

While the niche of a species is determined by a host of environmental, ecological and socio-economic factors of unknown influence and interaction strength, it is possible to exclude parts of the niche if the direct effects of one factor on a step rate-limiting to population persistence are well known. One such example for mosquito population persistence is whether temperature permits adult females to survive long enough to complete their first gonotrophic cycle and thus oviposit. Both adult female longevity and length of first gonotrophic cycle are temperature dependent. Combining these two relationships with a dynamic population-level simulation, *Brady et al. (2013*, *2014)* evaluated the thermal limits to persistence of *Ae. aegypti* and *Ae. albopictus* populations on a global scale. The binary outputs of this model are used as a mask to sample pseudo-absence points in locations known to be unsuitable—thereby informing the statistical model using mechanistic model outputs. The temperature suitability index developed by Brady et al. is also used in a continuous variable form (i.e., the relative number of ovipositions of parous females permitted by temperature) as a covariate in the BRT model.

## Occurrence records

The database used for this study contains information on the known global occurrences of the adults, pupae, larvae or eggs of *Ae. aegypti* and *Ae. albopictus* globally from 1960–2014. We included data from a variety of sources, including (1) published literature and (2) primary and unpublished occurrence data from national and international entomological surveys. To our knowledge this is the largest, most comprehensive global dataset for both *Ae. aegypti* and *Ae. albopictus.* Confirmed *Aedes* occurrences were entered in the database after a comprehensive literature search using methods described elsewhere (*Kraemer et al., 2015a*; *Kraemer et al., 2015b*; http://dx.doi.org/10.5061/dryad.47v3c). In short, this included extracting all available location (latitude and longitude) information from the relevant articles, primarily using Google Maps (http://www.google.com/maps) so that it matched the spatial resolution of our covariate datasets of approximately 5 km × 5 km. Primary and unpublished data sources were obtained from Brazil, Europe, Indonesia, Taiwan, and the United States. After consolidating all data into two large databases for each species, independently they underwent spatial and temporal standardization. An occurrence record was defined as a single occurrence at a given unique location within one calendar year. This was important to avoid over-representation in regions where multiple surveys per year were performed, such as Taiwan or Brazil. To ensure the accuracy of the data we overlaid the geolocated occurrence points with a raster that distinguished land from water. Any records that were positioned outside the land area were subsequently removed. In total we assembled 19,930 and 22,137 occurrence records for *Ae. aegypti* and *Ae. albopictus* respectively. The distribution of occurrence points are plotted in *Figure 1—figure supplement 4*.

## Land-cover and environmental variables

The distribution of both species considered in this study are known to be influenced by environmental factors such as temperature and demographic factors such as urbanisation (*Lounibos, 2002*; *Brown et al., 2014*). Global gridded maps of such variables are becoming ever more available and have been commonly applied in SDM and disease mapping (*Hijmans et al., 2005*; *Hay et al., 2006*; *Gething et al., 2011*; *Bhatt et al., 2013*; *Pigott et al., 2014a*, *2014b*). The rationale for the inclusion of each variable we used is described below.

### EVI

Survival of *Ae. aegypti* and *Ae. albopictus* is highly dependent on temperature and water availability (*Luz et al., 2008*). EVI measures vegetation canopy greenness and can be used as a proxy for soil surface-level moisture that are associated with the availability of mosquito larval development sites (*Estallo et al., 2008*; *Nihei et al., 2014*). Eggs and adults require moisture to survive, with low dry season moisture levels affecting adult mortality (*Sota and Mogi, 1992*; *Russell et al., 2001*). Vegetation canopy cover reduces evaporation and wind speed in the sub-canopy, which protects mosquito development sites (*Linthicum, 1999*; *Fuller et al., 2009*; *Hahn et al., 2014*). We used range and mean values of MODIS EVI after processing through a gap-filling algorithm described elsewhere (*Weiss et al., 2014*).

### Precipitation

The principal larval habitats of both species are man-made containers that are used for water storage or accumulate rain (*Morrison et al., 2004*). Some local studies have shown that there is a relationship between precipitation and vector abundance (*Scott et al., 2000*; *Romero-Vivas and Falconar, 2005*). To account for the availability of water-filled containers a maximum and minimum annual precipitation layer was extracted from the WorldClim database and projected for the year 2015 (http://www.worldclim.org).

### Urbanisation

*Ae. aegypti* adults are highly domesticated mosquitoes feeding almost exclusively on humans (*Bargielowski et al., 2013*), larvae develop preferentially in artificial containers in close association with human habitation, often in urban settings (*Lounibos, 2002*; *Honório et al., 2003*; *Brown et al., 2011*, *2014*; *Powell and Tabachnick, 2013*). *Ae. albopictus* are more commonly found in rural and peri-urban settings, feeding readily on a variety of mammalian and avian species, although *Ae. albopictus* shows similar larval development behavior in artificial containers (*Reiter, 2001*; *Gratz, 2004*; *Juliano and Philip Lounibos, 2005*; *Li et al., 2014*). To account for differences in urban, peri-urban and rural environments we built a categorical variable by supplementing the projected 2010 Global Rural Urban Mapping Project (GRUMP) urban and rural categories with land-cover classes using night-time light satellite imagery and population density, using the most up-to-date national censuses available to the

smallest available administrative unit available (*Balk et al., 2006*). A gridded surface of 5 km × 5 km cells was generated with each pixel representing either urban, peri-urban, or rural areas.

## Modelling approach

BRT models consistently outperform other species distribution models such as maximum entropy (Maxent), GARP, and BIOCLIM in their predictive performance (*Elith et al., 2006*; *Leathwick et al., 2006*). BRT combines the strengths of regression trees (i.e., the omission of irrelevant variables and the ability to model complex interactions) with machine learning techniques (i.e., the building of an ensemble of models that approximate the true response surface [*Elith and Leathwick, 2009*]). To prevent overfitting, the model used a penalized forward stepwise search and cross-validation method to identify the optimal number of decision trees (*Elith et al., 2008*). Modelling was performed using the gbm, dismo, raster and seegSDM R packages using the R v 3.1.1 environment (*Ridgeway, 2013*; *Golding, 2014*; *Hijmans, 2014*; *R Core Team, 2014*).

### Removing sample selection bias

Pseudo-absence (also referred to as background) records provide a sample of the set of conditions available to the species in the region rather than actual absences (*Phillips et al., 2009*). These records are needed because true absences are generally unavailable in large composite datasets such as the one used in this study. To account for reporting bias in presence data, a common problem with presence-only SDM, which if not accounted for can lead to biases in the resulting predictions, we follow *Phillips et al. (2009)* in sampling pseudo-absence points according to the same reporting bias likely to be present in occurrence records (namely spatial variation in reporting of mosquito occurrence). Firstly, we selected 10,000 occurrence records of *Aedes* species from the Global Biodiversity Information Facility (http://www.gbif.org), omitting all records of *Ae. aegypti* and *Ae. albopictus*. This dataset is intended to reflect biases in mosquito reporting in areas which are suitable for *Aedes* mosquitoes. Secondly, to reflect areas where habitats are biologically not suitable for *Aedes* occurrence we sampled an additional 10,000 pseudo-absence points at random locations, with sampling probability greatest in areas that the biologically-based temperature suitability index predicted to be unsuitable. Thirdly, sampling of occurrence points was also biased towards oversampled regions such as Brazil and Taiwan in which there were a large number of reported occurrence records due to the inclusion of results of large national entomological surveys (*Table 1*). Therefore, we weighted occurrence records from these locations so that the density of occurrence records per country matched the density of all other records globally by dividing the number of occurrence points by the size of the respective countries.

### Modelling

An ensemble BRT was constructed using 120 sub-models to derive uncertainty distributions of the prediction map. Each of the 120 sub-models was fitted to a separate bootstrap resampling of the dataset and used to generate a probability map for each individual species on a 5 km × 5 km resolution. The mean of these 120 sub-models was used as the final *Aedes* risk maps. Pixel based uncertainty was estimated by calculating the 95% confidence interval from the 120 sub-models.

### Predictive performance and relative influence of covariates

The variables used as land-cover and environmental correlates used in this study are quantified based on their relative influence (0–100) on explaining the variance in the models calculated as the sum of the number of times a particular variable is selected for splitting the decision tree, weighted by the squared improvement to the overall model averaged over all trees (*Friedman, 2001*; *Friedman and Meulman, 2003*). Note that in a BRT, non-informative predictors are largely ignored (*Elith et al., 2008*). Predictive performance of each sub-model was evaluated using the area under curve (AUC) statistic calculated as the mean AUC for each of the ten cross-validation folds evaluated against the other 90% of the data under the pairwise distance sampling procedure of *Hijmans (2012)*. The overall predictive accuracy of the model was measured as the mean and standard deviation of these AUCs across all 120 sub-models (*Merckx et al., 2010*; *Hijmans, 2012*).

## Acknowledgements

MUGK is funded by the German Academic Exchange Service (DAAD) through a graduate scholarship. MES is funded by a project grant from the Bill & Melinda Gates Foundation as part of the VecNet consortium (http://vecnet.org). FMS is funded by the Rhodes Trust. CMB acknowledges funding from

the U.S. National Aeronautics and Space Administration (#NNX15AF36G). CGM was funded in part by contract N01-A1-25489 from the NIH/National Institute of Allergy and Infectious Diseases. IRFE is funded by the Wellcome Trust (#099872). WVB, GH, and FS acknowledge funding from VBORNET, an ECDC funded project (contract number ECDC/09/018). OJB is funded by a BBSRC studentship. JPM is funded by, and SIH, GRWW, TWS and OJB acknowledge the support of the International Research Consortium on Dengue Risk Assessment Management and Surveillance (IDAMS, European Commission seventh Framework Programme (#21803) http://www.idams.eu). The contents of this publication are the sole responsibility of the authors and do not necessarily reflect the views of the European Commission. DMP is funded by a Sir Richard Southwood Graduate Scholarship from the Department of Zoology at the University of Oxford. TWS acknowledges funding from the Bill & Melinda Gates Foundation (#OPP52250), the Innovative Vector Control Consortium, and the NIH (R01-AI069341, R01-AI091980, R01-GM08322, and 1P01AI098670). TWS, CMB, DLS and SIH also acknowledge funding support from the Research and Policy in Infectious Diseases Dynamics (RAPIDD) program of the Science and Technology Directorate, Department of Homeland Security, and the Fogarty International Center, National Institutes of Health. NG is funded by a grant from the Bill & Melinda Gates Foundation (#OPP1053338). SIH is funded by a Senior Research Fellowship from the Wellcome Trust (#095066), which also supports KAD and AQNM.

We want to thank Ralph E Harbach for his comments on the final manuscript. We thank Dr Roseli La Corte dos Santos, Departamento de Morfologia, Universidade Federal de Sergipe for providing useful guidance, the Ministry of Health of Brazil municipality secretaries and Centers for Disease Control of Taiwan for providing mosquito occurrence data and Bimandra Djaafara and Karin Dian Lestari of EOCRU for their work on geo-referencing the Indonesia *Aedes* datasets. We also want to thank the VBORNET & TigerMaps network funded by ECDC, Stockholm, and all their contributors (a detailed list is given in *Supplementary file 1*) for releasing their vector distribution data. The funders had no role in study design, data collection and analysis, decision to publish, or preparation of the manuscript.

## Additional information

### Competing interests

SIH: Reviewing editor, *eLife.* The other authors declare that no competing interests exist.

### Funding

| Funder | Grant reference | Author |
| --- | --- | --- |
| Studienstiftung des Deutschen Volkes | | Moritz UG Kraemer |
| Bill and Melinda Gates Foundation | #OPP1053338 | Nick Golding |
| Wellcome Trust | #095066 | Kirsten A Duda, Adrian QN Mylne, Simon I Hay |
| European Centre for Disease Prevention and Control | ECDC/09/018 | Wim Van Bortel, Guy Hendrickx, Francis Schaffner |
| European Commission Directorate-General for Research and Innovation | #21803 | Oliver J Brady, Jane P Messina |
| Biotechnology and Biological Sciences Research Council (BBSRC) | | Oliver J Brady |
| National Aeronautics and Space Administration (NASA) | #NNX15AF36G | Christopher M Barker |
| National Institutes of Health (NIH) | RAPIDD program | Christopher M Barker, Thomas W Scott, David L Smith, Simon I Hay |
| National Institutes of Health (NIH) | R01-AI069341 | Thomas W Scott |
| National Institutes of Health (NIH) | R01-AI091980 | Thomas W Scott |

| Funder | Grant reference | Author |
|---|---|---|
| National Institutes of Health (NIH) | R01-GM08322 | Thomas W Scott |
| National Institutes of Health (NIH) | N01-A1-25489 | Chester G Moore |
| Bill and Melinda Gates Foundation | #OPP52250 | Thomas W Scott |
| Sir Richard Southwood Graduate Scholarship | | David M Pigott |
| Wellcome Trust | Vecnet | Marianne E Sinka |
| Wellcome Trust | #099872 | Iqbal RF Elyazar |
| The Rhodes Trust | | Freya M Shearer |

The funders had no role in study design, data collection and interpretation, or the decision to submit the work for publication.

## Author contributions

MUGK, Conception and design, Acquisition of data, Analysis and interpretation of data, Drafting or revising the article; MES, FMS, Acquisition of data, Drafting or revising the article; KAD, AQNM, Acquisition of data; CMB, CGM, RGC, GEC, WVB, FS, IRFE, H-JT, Drafting or revising the article, Contributed unpublished essential data or reagents; GH, Contributed unpublished essential data or reagents; OJB, NG, SIH, Conception and design, Analysis and interpretation of data, Drafting or revising the article; JPM, DMP, DLS, Analysis and interpretation of data, Drafting or revising the article; TWS, Conception and design, Drafting or revising the article; GRWW, Analysis and interpretation of data, Drafting or revising the article, Contributed unpublished essential data or reagents

## Author ORCIDs

Christopher M Barker, http://orcid.org/0000-0002-7941-346X
Francis Schaffner, http://orcid.org/0000-0001-9166-7617
David M Pigott, http://orcid.org/0000-0002-6731-4034
David L Smith, http://orcid.org/0000-0003-4367-3849
Nick Golding, http://orcid.org/0000-0001-8916-5570
Simon I Hay, http://orcid.org/0000-0002-0611-7272

# Additional files

## Supplementary file

• Supplementary file 1. List of contributors and their affiliation from TigerMaps & VBORNET for *Ae. albopictus* presence records in Europe.

## Major dataset

The following dataset was generated:

| Author(s) | Year | Dataset title | Dataset ID and/or URL | Database, license, and accessibility information |
|---|---|---|---|---|
| Kraemer MUG, Sinka ME, Duda KA, Mylne A, Shearer FM, Brady OJ, Messina JP, Barker CM, Moore CG, Carvalho RG, Coelho GE, Van Bortel W, Hendrickx G, Schaffner F, Wint GRW, Elyazar IRF, Teng H, Hay SI | 2015 | Data from: The global compendium of Aedes aegypti and Ae. albopictus occurrence | http://dx.doi.org/10.5061/dryad.47v3c | Available at Dryad Digital Repository under a CC0 Public Domain Dedication. |

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
