## [Decision Letter]

Thank you for submitting your work entitled “The global distribution of the arbovirus vectors *Aedes aegypti* and *Ae. albopictus*” for peer review at *eLif*e. Your submission has been favorably evaluated by Prabhat Jha (Senior editor), a Reviewing editor, and two reviewers.

The reviewers have discussed the reviews with one another and the Reviewing editor has drafted this decision to help you prepare a revised submission.

The following individuals responsible for the peer review of your submission have agreed to reveal their identity: Mark Jit (Reviewing editor) and Louis Lambrechts (peer reviewer). A further reviewer remains anonymous.

Summary:

The manuscript provides a comprehensive survey of the documented geographical distribution of *Aedes aegypti* and *Aedes albopictus* mosquitoes. The occurrence data are used to establish a predictive map of global distribution based on a set of relevant environmental variables.

Overall, the reviewers felt that your work provided a valuable synthesis of important data related to a key global health problem, the distribution one of the most important disease vectors. They felt that the paper was remarkably clear and well written, apart from a few key areas which would benefit from improved clarity.

Essential revisions:

1) It is claimed that one of the main advances here is the inclusion of “socio-economic” data. However, that data are limited to a single fairly complex variable, urbanisation. Urbanness is more of a demographic factor and does not necessarily do anything to distinguish socio-economic status; any city has both poor and rich areas, both of which are equally “urban”. The inclusion of this variable is important, but it should be clarified that it does not actually distinguish socio-economic status.

Related to the above, the last sentence in the fourth paragraph of the Introduction should be revised. “Ignoring socio-economic factors” should be softened given the fact that the current manuscript only does so to a very limited degree and it is clearly a major challenge to do so more in depth at a global scale. Also, “poorly distributed input occurrence data” is vague. Most occurrence data—including that in this manuscript—is poorly distributed.

2) There should be more discussion about some of the findings. What does the analysis tell us about the potential importance of other variables? Where are the knowledge gaps? Was there any insight into habitat differences between *Ae. aegypti* and *Ae. albopictus*? What are the hypothesized biological effects of EVI range? How much does this improve upon the Brady suitability model? It would be interesting to explore whether the combined global distributions of both mosquito species could provide any insights about the conditions leading to sympatric co-existence or competitive exclusion.

---

## [Author Response]

*1) It is claimed that one of the main advances here is the inclusion of “socio-economic” data. However, that data are limited to a single fairly complex variable, urbanisation. Urbanness is more of a demographic factor and does not necessarily do anything to distinguish socio-economic status; any city has both poor and rich areas, both of which are equally “urban”. The inclusion of this variable is important, but it should be clarified that it does not actually distinguish socio-economic status*.

Thank you for requesting clarification on the inclusion of the urbanisation variable. We appreciate that urbanisation can be considered a land-cover and/or demographic variable. Therefore we changed the phrasing throughout the manuscript to ‘land-cover variable’ and ‘demographic information’. In addition to this we have removed any claims that we have included ‘socio-economic’ covariates.

*Related to the above, the last sentence in the fourth paragraph of the Introduction should be revised. “Ignoring socio-economic factors” should be softened given the fact that the current manuscript only does so to a very limited degree and it is clearly a major challenge to do so more in depth at a global scale. Also, “poorly distributed input occurrence data” is vague. Most occurrence data—including that in this manuscript—is poorly distributed*.

Following your suggestion we softened the phrasing of the sentence replacing “ignoring socio-economic factors’ with ‘exclusively focusing on meteorological factors’. We believe that this more accurately reflects how previous studies have predicted the environmental niche of *Ae. aegypti* and *Ae. albopictus* methodologically*.* To address the second point raised regarding the ‘poorly distributed’ occurrence data we replaced this phrase by ‘many models used small sets of input occurrence data’.

*2) There should be more discussion about some of the findings. What does the analysis tell us about the potential importance of other variables? Where are the knowledge gaps? Was there any insight into habitat differences between* Ae. aegypti *and* Ae. albopictus*? What are the hypothesized biological effects of EVI range? How much does this improve upon the Brady suitability model? It would be interesting to explore whether the combined global distributions of both mosquito species could provide any insights about the conditions leading to sympatric co-existence or competitive exclusion*.

The reviewer makes an important point as our analysis does give some important insights into the habitat differences between the two species. The main difference, which we have now highlighted in the manuscript, is the increased preference for urban and peri-urban areas for *Ae. aegypti* (see the relative influence plots in Figure 1—figure supplement 1)*,* this is supported by the increased influence of mean EVI in mid-ranging values for *Ae. albopictus*. These relative influence plots as well as the distribution of the occurrence points support the hypothesis that, on average, *Ae. albopictus* is more frequently observed in habitats characterized by vegetation rather than in and around human habitats, whereas the converse is true for *Ae. aegypti* (see Discussion, second paragraph). This does not, however, rule out the possibility that the two species can overlap. Further finer scale studies need to be conducted to investigate if competitive exclusion for hosts and/or habitat occurs between *Ae. aegypti* and *Ae. albopictus*.

With regards to the developments of this analysis over the Brady et al*.* suitability paper, the later was focussed exclusively on the effects of temperature on limiting the global extent of *Ae. aegypti* and *Ae. albopictus* persistence and ability to transmit dengue virus. The current analysis incorporates additional covariates (such as precipitation and urban areas) to account for additional variance in the data. This is shown by the fact that the temperature suitability layer from Brady et al. only accounts for half of variation as demonstrated by the relative influence plots.